# MRI bias field correction with an implicitly trained CNN.

**Attila Simkó**[1]                                                          ATTILA.SIMKO@UMU.SE
[1] *Department of Radiation Sciences, Umeå University, Umeå, Sweden*
**Tommy Löfstedt**[2]
[2] *Department of Computing Science, Umeå University, Umeå, Sweden*

**Anders Garpebring**[1]
**Tufve Nyholm**[1]
**Joakim Jonsson**[1]

## Abstract

In magnetic resonance imaging (MRI), bias fields are difficult to correct since they are inherently unknown. They cause intra-volume intensity inhomogeneities which limit the performance of subsequent automatic medical imaging tasks, *e.g.*, tissue-based segmentation. Since the ground truth is unavailable, training a supervised machine learning solution requires approximating the bias fields, which limits the resulting method.

We introduce implicit training which sidesteps the inherent lack of data and allows the training of machine learning solutions without ground truth. We describe how training a model implicitly for bias field correction allows using non-medical data for training, achieving a highly generalized model. The implicit approach was compared to a more traditional training based on medical data. Both models were compared to an optimized N4ITK method, with evaluations on six datasets.

The implicitly trained model improved the homogeneity of all encountered medical data, and it generalized better for a range of anatomies, than the model trained traditionally. The model achieves a significant speed-up over an optimized N4ITK method—by a factor of 100, and after training, it also requires no parameters to tune.

For tasks such as bias field correction—where ground truth is generally not available, but the characteristics of the corruption are known—implicit training promises to be a fruitful alternative for highly generalized solutions.

**Keywords:** Self-supervised learning, Implicit Training, Magnetic Resonance Imaging, Bias Field Correction, Image Restoration

## 1. Introduction

Despite the topicality and recent successes of Artificial Neural Networks (ANNs) in deep learning, the lack of available data is still a limitation in many areas of research. A supervised machine learning solution inherently requires corresponding input and target data. However, in image restoration, for example, the data acquisition process might only provide observations that can not be separated into input and target. In such cases, to train a neural network, the target can only be an approximation of the ground truth. The focus then becomes how well these approximations hold for a wide range of data.

For ANNs to be relevant for medical imaging applications, one must consider the performance for observations across different scanners and scanner parameters, and the scanned regions of the body. Concerns about overfitting need to be addressed when collecting data

for training. Confronting the issue of inseparable observations with generalizability in mind, we propose a novel, simple training technique for ANNs, which only requires general knowledge about the corruption and how it operates. The technique is denoted *implicit training*.

Imperfections during the image acquisition process in MRI and the patient anatomy collectively creates a multiplicative corruption, called a bias field. The field can cause a 10–40 % intra-volume smooth, low-frequency intensity variation (Meyer et al., 1995; Sled et al., 1997). Due to the bias field, the signal intensity of homogeneous tissue from MRI data is seldom homogeneous. Although having limited impact on visual diagnosis, the bias field has crucial implications on automated downstream processing. Information about the current state of bias field correction can be found in the appendix.

The main contribution of the presented study is the implicit training approach. The paper details, implements and tests the generalizability of a model trained in such a manner, against a model trained more conventionally. This investigation results in a bias field correction model trained on non-medical images[1], which achieves comparable results to the widely used N4ITK method, with a speed-up factor of 100.

## 2. Materials and Methods

In image restoration, the general purpose is to recover the latent, clear image, $\mathbf{u}$, from a corrupted observation, $\mathbf{v} = H(\mathbf{u}, \mathbf{b})$, with $H$ as the corruption function applying the corruption, $\mathbf{b}$, to the image, $\mathbf{u}$. Since bias fields are multiplicative, the observation is

$$\mathbf{v} = H(\mathbf{u}, \mathbf{b}) = \mathbf{u} \odot \mathbf{b}, \tag{1}$$

where the corruption function $H$ is the Hadamard product ($\odot$, element-wise multiplication) between the clear image and the corruption. The true signal intensity $\mathbf{u}$ only contains intensity variations of relevance, and $\mathbf{b}$ is the bias field. We describe two ways to train a machine learning model that learns $\mathcal{F} : \mathbb{R}^{256 \times 256} \to \mathbb{R}^{256 \times 256}$, such that

$$\mathbf{b} \approx \widehat{\mathbf{b}} = \mathcal{F}(\mathbf{v}). \tag{2}$$

With the aim of adhering to the model definition in Eq. (2), we are interested in finding a function $\mathcal{F}$, that minimises the mean squared error,

$$\mathcal{L}(\mathbf{v}, \mathbf{b}) = \frac{1}{n} \big\| \mathcal{F}(\mathbf{v}) - \mathbf{b} \big\|_2^2, \tag{3}$$

where $n$ is the number of voxels in $\mathbf{v}$.

To further encourage the networks to perform the intended task, an identity regularization term $R$ was added to ensure consistency. After correcting $\mathbf{v}$, the corrected image should be bias-free, *i.e.* correcting an image a second time should simply return an identity bias field. The regularization term is defined as

$$\frac{1}{n}\Big\|\mathcal{F}\Big(H^{-1}\big(\mathbf{v}, \mathcal{F}(\mathbf{v})\big)\Big) - \mathbb{1}\Big\|_2^2 \approx \frac{1}{n}\Big\|\mathcal{F}\Big(H^{-1}\big(H(\widehat{\mathbf{u}}, \widehat{\mathbf{b}}), \widehat{\mathbf{b}}\big)\Big) - \mathbb{1}\Big\|_2^2 = \frac{1}{n}\big\|\mathcal{F}(\widehat{\mathbf{u}}) - \mathbb{1}\big\|_2^2 =: R(\mathbf{v}),$$

---

1. The model is available online: https://doi.org/10.5281/zenodo.3749526

where $\widehat{\mathbf{u}}$ is the approximated bias-free image, $\mathbb{1}$ is a $256 \times 256$ image of ones, and $H^{-1}$ is the inverse of the corruption function $H$, such that $H^{-1}(H(\mathbf{v}, \mathbf{b}), \mathbf{b}) = \mathbf{v}$. For the case of bias fields, the inverse function is $H^{-1}(\mathbf{v}, \mathbf{b}) = \mathbf{v} \oslash \mathbf{b}$, element-wise division.

From the formulation of the problem, we know that $\mathbf{b}$ is not available, but it is required to compute the loss. This is often a concern for other image restoration tasks as well, and it is the motivation for this work. For these cases there are several approaches to train a model explicitly, *i.e.* by keeping this loss function. For example we can approximate $\mathbf{b}$ by applying a reliable analytical correction method on $\mathbf{v}$. We have taken another approach to train explicitly.

### 2.1. Bias Field Generator

We start by generating bias fields that follow the characteristics from literature (Zujun, 2006; Vovk et al., 2007), their description (Sled et al., 1998) and the examples from the BrainWeb website.

The method used to generate 2D spatial random but physically plausible fields followed the description (Heße et al., 2014). For a single bias field, it used a Gaussian covariance model in the form

$$\mathrm{cov}(r) = \exp\left(-\frac{\pi}{4} \cdot \left(\frac{r}{\ell}\right)^2\right), \tag{4}$$

where $r$ is the distance from a randomly chosen peak of the Gaussian, and $\ell$ is an arbitrary length scale corresponding to the frequency of the generated bias fields. A lower $\ell$ corresponds to a higher frequency. We selected $\ell$ for each bias field randomly in the range $[10, 50]$, based on visual comparison to the bias fields from BrainWeb. The field is then scaled to have a mean of 1 and an absolute maximum chosen randomly in the range $[1.1, 1.3]$, corresponding to a maximum inhomogeneity of 20–60 %, respectively.

### 2.2. Explicit Training

We constructed the training dataset by collecting bias-free data from BrainWeb for $\mathbf{u}$ while an in-house bias field generator described in Section 2.1 provided $\mathbf{b_g}$. Hence, the total explicit loss function was

$$\mathcal{L}_E(\mathbf{v}, \mathbf{b_g}) = \mathcal{L}(\mathbf{v}, \mathbf{b_g}) + \alpha R(\mathbf{v}), \tag{5}$$

where $\alpha$ is a regularization parameter.

The explicitly trained model, by design, favors images with the same modality and scanned region as the images in the training set, exposing the issue of generalizability. Note that methods to improve generalizability of the explicit training process do exist. Collecting a more diverse training dataset will lead to a more general model, however for real data, the target bias fields can only be approximated, and covering all anatomies equally is a difficult task. The authors therefore propose a training process that circumvents these difficulties.

### 2.3. Implicit Training

The *implicit training* process is an option as long as the characteristics of $\mathbf{b}$ are known, and therefore can appropriately be artificially generated, and the function $H$ is known and invertible.

The basis of the implicit training is a randomly generated artificial bias field, $\mathbf{b_g}$. Given an observation, $\mathbf{v}$, with an unknown bias field, $\mathbf{b}$, we can construct

$$\mathbf{v_g} = H(\mathbf{v}, \mathbf{b_g}) = H(H(\mathbf{u}, \mathbf{b}), \mathbf{b_g}).$$

For this image, $\mathbf{v_g}$, the underlying bias field is naturally $\mathbf{b} \odot \mathbf{b_g}$, which is still unknown. However, note that we want the model to learn to return the bias field of a corrupted image, such that $\mathcal{F}(\mathbf{v_g}) = H(\widehat{\mathbf{b}}, \widehat{\mathbf{b_g}}) \approx H(\mathbf{b}, \mathbf{b_g}) = \mathbf{b} \odot \mathbf{b_g}$ and that $\mathcal{F}(\mathbf{v}) = \widehat{\mathbf{b}} \approx \mathbf{b}$. Having an $\mathcal{F}$ adhering to this, we see that

$$H^{-1}(\mathcal{F}(\mathbf{v_g}), \mathcal{F}(\mathbf{v})) \,=\, H^{-1}(H(\widehat{\mathbf{b}}, \widehat{\mathbf{b_g}}), \widehat{\mathbf{b}}) \,=\, \widehat{\mathbf{b}} \odot \widehat{\mathbf{b_g}} \oslash \widehat{\mathbf{b}} \,=\, \widehat{\mathbf{b_g}} \approx \mathbf{b_g}.$$

Using this, we define an implicit loss function making the model learn not through its output, but through the relationship between two of its outputs. We thus define the implicit loss,

$$\frac{1}{n}\left\|H^{-1}\big(\mathcal{F}(\mathbf{v_g}), \mathcal{F}(\mathbf{v})\big) - \mathbf{b_g}\right\|_2^2 \,=\, \frac{1}{n}\left\|\mathcal{F}(\mathbf{v_g}) \oslash \mathcal{F}(\mathbf{v}) - \mathbf{b_g}\right\|_2^2 \,=:\, \mathcal{L}_I(\mathbf{v}, \mathbf{b_g}). \tag{6}$$

In every training iteration, we generate $\mathbf{b_g}$, construct $\mathbf{v_g} = H(\mathbf{v}, \mathbf{b_g})$, take the output of $\mathcal{F}$ for both $\mathbf{v_g}$ and $\mathbf{v}$, and fit their element-wise ratio to be $\mathbf{b_g}$. Implicit training for the input $\mathbf{v}$ with the simulated bias field $\mathbf{b_g}$ is illustrated in Fig. 1.

Implicit training excludes $\mathbf{b}$ from the loss function, so it can remain unknown, and introduces a term $\mathbf{b_g}$ which is a randomly generated bias field, following the statistics of $\mathbf{b}$. In essence, since no information that exists both in $\mathbf{v}$ and $\mathbf{v_g}$ is included in $\mathcal{L}_I$, we claim that by minimizing this loss, $\mathcal{F}$ learns not only to return $\mathbf{b_g}$, but also to return features that are indistinguishable from the characteristics of $\mathbf{b_g}$. Therefore, if we correctly generated $\mathbf{b_g}$ with the same characteristics as $\mathbf{b}$, training should also—implicitly—minimize $\mathcal{L}_E$. Together with regularization terms on both $\mathbf{v}$ and $\mathbf{v_g}$, the total implicit loss becomes

$$\mathcal{L}(\mathbf{v}, \mathbf{b_g}) = \mathcal{L}_I(\mathbf{v}, \mathbf{b_g}) + \alpha\Big(R(\mathbf{v}) + R(\mathbf{v_g})\Big), \tag{7}$$

where $\alpha$ is a regularization parameter.

Due to the training process, the training dataset is now not constrained to images where $\mathbf{b}$ is known and also the image content is not relevant. Therefore the illustrated process for the presented model was implemented without using medical data, by training on a subset of ImageNet images.

## 2.4. CNN Architecture

The CNN model, $\mathcal{F}$, for correction had an input and output size of $256 \times 256 \times 1$. The architecture was a U-Net (Ronneberger et al., 2015) with five down-sampling levels. Each down-sampling block contains two convolutional layers with symmetric padding and ReLU activations (Glorot et al., 2011), Batch Normalization (Ioffe and Szegedy, 2014) and for down-sampling to half the size, the model used Max Pooling layers. Each up-sampling layer is constructed similarly using a bilinear upsampling layer, and an additional Gaussian blur layer. The final output is constrained to values between 0.5 and 2.

## 3. Experiments

**Explicit**   To use the explicit loss in Eq. (7) bias-free images are needed. For that we used the Simulated Brain Database, available from BrainWeb[2] (Cocosoc et al., 2002; Kwan et al., 1999; Evans et al., 1996; Collins et al., 1998) and using the 20 distinct tissue maps, we simulate signal from five different MR contrasts using Matlab R2018b (The MathWorks, Inc., Natick, Massachusetts, United States). The images **u** were corrupted by a 5 % Gaussian noise for higher stability and they were corrupted using a generated bias field $\mathbf{b_g}$.

We generated 120 000 images, which were split between training the explicit model (108 000), validating the explicit and the implicit models (6 000) and for evaluating all methods (6 000). All images were saved with their corresponding tissue maps for gray matter (GM), white matter (WM), and cerebrospinal fluid (CSF).

**Implicit**   We trained another model using the implicit loss from Eq. (6), where by design, our options for the training data are less restricted. For training we used a random sample of 108 000 images—the same as the number of training images for the explicit approach—from ImageNet, the large-scale image collection containing objects from about 5 000 categories. The changes in lighting, by smooth spatial color gradients or fading can follow similar characteristics as bias fields, therefore these changes are considered the bias fields of natural images, **b**. For each image we also generated a bias field $\mathbf{b_g}$. We used the same validation dataset as for explicit training, 6 000 images from BrainWeb.

**N4ITK**   For all evaluations we corrected the bias field using N4ITK from the SimpleITK Python package[3]. N4ITK was used on images rescaled to $[0, 100]$, which improves the performance. The parameters were optimized for the given dataset using grid search.

---

2. http://brainweb.bic.mni.mcgill.ca
3. http://www.simpleitk.org

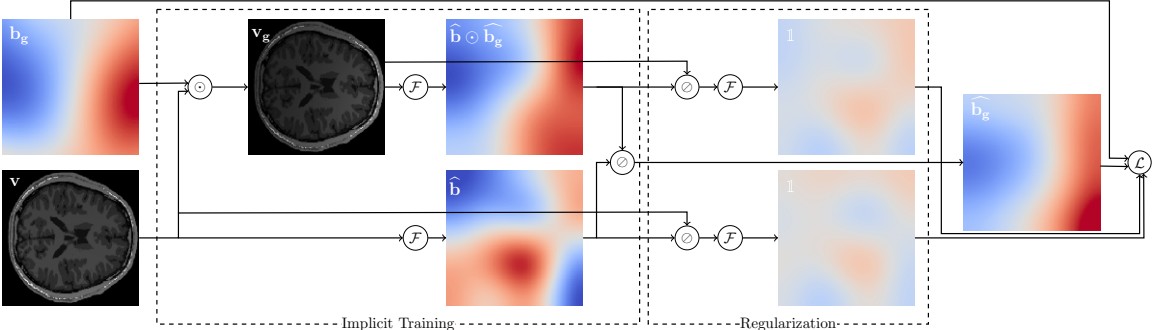

Figure 1: Implicit Training. The input is an image with an unknown bias field and another, generated bias field. The $\odot$ and $\oslash$ denote element-wise multiplication and division, respectively. The model is denoted $\mathcal{F}$, and the operations define $\mathcal{L}$ in Eq. (7). The illustrated bias fields are real outputs of the implicitly trained model.

### 3.1. Evaluation on BrainWeb

We evaluated the methods on the testing dataset of 6 000 synthetic images generated from BrainWeb. The evaluation metrics were the Coefficient of Variation (CV) and the Structural Similarity Index (SSIM). The experiment shows how homogeneous the signal intensity is for GM, WM, and CSF before and after correction.

We also evaluated the methods based on the execution time to correct all images in the dataset. To bring N4ITK and the proposed methods to a common ground, the timing experiments were performed on a single CPU. Note that the execution time of N4ITK can be drastically improved due to its straightforward slice-wise parallelism, but the same holds for the proposed models by increasing the batch size and the use of a GPU.

### 3.2. Evaluation on phantom data

A typical benchmark test for bias field correction is to apply the given method on data that contains a homogeneous object with a simple geometry, assuming an homogeneous signal. For this test, 150 $T_2$-weighted slices were acquired with a 3T Signa PET/MR scanner (GE Healthcare, Chicago, Illinois, United States) at the University Hospital of Umeå, Umeå, Sweden, of a spherical phantom object. The homogeneity of the object means that all non-uniformity inside the volume was bias only, simplifying the problem to

$$\mathcal{F}(\mathbf{v}) = \mathcal{F}(\mathbf{b}) = \widehat{\mathbf{b}}. \tag{8}$$

The corrections were evaluated in MICE Toolkit[4] (Nyholm and Jonsson, 2014) for all slices based on the CV calculated for the pixels inside the phantom.

### 3.3. Evaluation on real data

Two brain and two pelvic scans using different contrasts were collected at the University Hospital of Umeå, Umeå, Sweden, each with two segmented tissues.

One of the brain scans was a $T_1$-weighted gradient echo sequence with segmented tissues of white matter (WM) on 71 slices and connective tissue of the scalp (SC) on 101 slices. The other scan was a $T_2$-weighted scan with the same segmented tissues, WM on 52 and SC on 42 slices. One of the pelvic scans was a LAVA-FLEX sequence and the segmented tissues were fat (F) on 40 and the bladder (B) on 14 slices. The other scan was a $T_1$-weighted spin-echo and the selected tissues were fat (F) on 101 and muscle (M) on 115 slices.

The corrections were evaluated in MICE Toolkit.

## 4. Results

Both explicit and implicit training methods went through hyper-parameter tuning, and the best CV results on the validation dataset were achieved using the NAdam optimizer (Dozat, 2016) using a batch size of 32. The explicit training method achieved the best results after 134 epochs using a learning rate of 0.005 and $\alpha = 0.5$, while the implicit training took 161 epochs using a learning rate 0.0001 with $\alpha = 0.2$.

---

4. NONPI Medical AB, Umeå, Sweden; Website: https://www.micetoolkit.com

Table 1: Results of the BrainWeb dataset, with the best for each tissue in bold. (Multiple, if the Nemenyi test shows no significant differences.)

| | CV | | | SSIM | | | Time [s] |
|---|---|---|---|---|---|---|---|
| | *GM* | *WM* | *CSF* | *GM* | *WM* | *CSF* | |
| *Original* | $0.083 \pm 0.007$ | $0.047 \pm 0.006$ | $0.250 \pm 0.019$ | $1.000 \pm 0.000$ | $1.000 \pm 0.000$ | $1.000 \pm 0.000$ | |
| *Corrupted* | $0.196 \pm 0.079$ | $0.165 \pm 0.078$ | $0.327 \pm 0.065$ | $0.927 \pm 0.067$ | $0.918 \pm 0.074$ | $0.929 \pm 0.061$ | |
| *N4ITK* | $\mathbf{0.128 \pm 0.063}$ | $\mathbf{0.089 \pm 0.061}$ | $\mathbf{0.284 \pm 0.049}$ | $0.970 \pm 0.047$ | $0.966 \pm 0.053$ | $\mathbf{0.971 \pm 0.044}$ | 25657 |
| *Explicit* | $0.152 \pm 0.057$ | $0.121 \pm 0.062$ | $0.291 \pm 0.047$ | $0.970 \pm 0.029$ | $0.963 \pm 0.037$ | $0.969 \pm 0.028$ | **232** |
| *Implicit* | $0.144 \pm 0.047$ | $0.111 \pm 0.052$ | $0.287 \pm 0.036$ | $\mathbf{0.972 \pm 0.019}$ | $\mathbf{0.967 \pm 0.025}$ | $\mathbf{0.971 \pm 0.019}$ | **232** |

For the **BrainWeb** results see Table 1. N4ITK was optimized for $2\times$ downsampling and 6 control points, which results in much higher frequency changes than the ground truth bias (as seen on Fig. 3). The results were compared for significance using a Friedman test of equivalence followed by a Nemenyi post-hoc test (Demšar, 2006) with a threshold of 0.05.

Table 2 collects the results for the **Phantom** and the **Real** data. The performance of the explicit method suggests that it doesn't generalize well to real data. This is further supported by the high standard deviation, visual assessment on Fig. 2 and in the appendix (on Fig. 4). For N4ITK the grid search method did not find a set of parameters that would improve the CV of the phantom data, and most configurations introduced a defect to the image. For this data, the images were scaled between $[0, 5\,000]$, having 6 control points without downsampling. For the $T_1$-weighted and $T_2$-weighted brain scans, and the LAVA-FLEX and $T_1$-weighted pelvic scans were optimized for 8, 6, 4 and 8 control points and $2\times$, $2\times$, $2\times$ and $1\times$ downsampling, respectively. (See examples on Fig. 4.)

## 5. Discussion and Conclusions

Both trained models performed similarly to N4ITK on the BrainWeb dataset, and while CV favored N4ITK, it was outperformed by the implicit method for SSIM. Their similar performance is further shown in the appendix (on Fig. 3). Although the explicit method has been trained on similar BrainWeb scans, the implicit approach still achieved better results. Their difference in performance becomes more significant for the other evaluation datasets. This showcases the opportunities of implicit training, which by design produces highly generalizable models, which is crucial for reliable machine learning methods.

Our second contribution is a fast and efficient machine learning method for bias field correction, which challenges the manually-tuned, state-of-the-art N4ITK with a $100\times$ speed-

| Original | N4ITK | Explicit | Implicit |
|---|---|---|---|

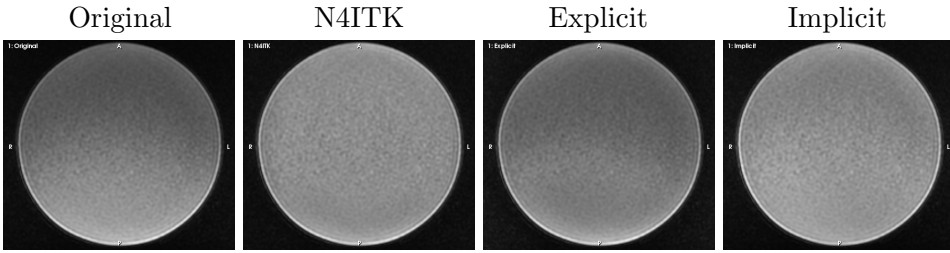

Figure 2: Visual comparison of the results for an example slice of the phantom dataset.

Table 2: CV results of the phantom and real data, with the best for each tissue in bold. (Multiple, if the Nemenyi test shows no significant differences.)

| Dataset | Contrast | Tissue | *Corrupted* | *N4ITK* | *Explicit* | *Implicit* |
|---|---|---|---|---|---|---|
| *Phantom* | | | $0.214 \pm 0.028$ | $\mathbf{0.054 \pm 0.004}$ | $0.157 \pm 0.014$ | $0.090 \pm 0.010$ |
| *Brain MRI* | $T_1$-weighted | *WM* | $0.136 \pm 0.022$ | $\mathbf{0.098 \pm 0.015}$ | $0.112 \pm 0.018$ | $\mathbf{0.100 \pm 0.012}$ |
| | | *SC* | $0.237 \pm 0.078$ | $0.197 \pm 0.042$ | $0.212 \pm 0.046$ | $\mathbf{0.122 \pm 0.019}$ |
| | $T_2$-weighted | *WM* | $0.149 \pm 0.022$ | $\mathbf{0.123 \pm 0.018}$ | $0.161 \pm 0.026$ | $0.138 \pm 0.017$ |
| | | *SC* | $0.177 \pm 0.010$ | $0.172 \pm 0.010$ | $0.169 \pm 0.012$ | $\mathbf{0.162 \pm 0.008}$ |
| *Pelvic MRI* | LAVA-FLEX | *F* | $0.130 \pm 0.033$ | $\mathbf{0.115 \pm 0.029}$ | $0.153 \pm 0.041$ | $\mathbf{0.109 \pm 0.025}$ |
| | | *B* | $0.154 \pm 0.012$ | $0.149 \pm 0.010$ | $0.151 \pm 0.013$ | $\mathbf{0.140 \pm 0.008}$ |
| | $T_1$-weighted | *F* | $0.174 \pm 0.018$ | $\mathbf{0.129 \pm 0.017}$ | $0.166 \pm 0.020$ | $\mathbf{0.131 \pm 0.012}$ |
| | | *M* | $0.205 \pm 0.008$ | $\mathbf{0.193 \pm 0.008}$ | $0.205 \pm 0.007$ | $\mathbf{0.189 \pm 0.006}$ |

up factor. The trained model improves the homogeneity of all encountered datasets both by quantitative evaluation and visual assessment. Our method requires no parameters to tune at evaluation, which should prove most useful when manual tuning is not available.

Using N4ITK we have also encountered an artefact for the $T_1$-weighted pelvic scan for all sets of parameters, that distinctly separates the fat area into the darker and brighter regions. This artefact was not present in either the explicitly or the implicitly trained results as by design they can only return a smooth bias field.

Results in Table 2 show that N4ITK performed significantly better for the phantom dataset, and one tissue, while the implicit model achieved significantly better results for three tissue evaluations. For the four remaining tissues N4ITK and the implicit model performed without significant differences. The corrections visualized orthogonal to the correction axis in the appendix (on Fig. 4) show artefacts of the explicit model due to the 2D nature of the correction method. No such artefacts are present for the implicitly trained model. In fact the LAVA-FLEX dataset contained darker and brighter slices especially around the edges of the image, and the effect of these artefacts were even reduced by the implicit model. For the Phantom dataset, the 3D N4ITK method achieved the lowest standard deviation, yet for all other real data, this was the lowest for the implicit model. The comparable standard deviations to a 3D method, and visual evaluations show that although the problem is 3D, meaningful bias field correction can be achieved by a 2D approach.

The results and easy implementation suggest that the proposed implicit training process could be used for other tasks as well. The two requirements are that the image corruption can be simulated accurately, and that the relation of the image and the corruption is known, and is invertible. Implicit training could prove useful for further tasks, such as other types of image corruption (Gaussian denoising, MRI motion artefact correction), image impainting or even object detection, by allowing to train on a wider range of data.

## Acknowledgments

We are grateful for the financial support obtained from the Cancer Research Foundation in Northern Sweden (LP 18-2182, AMP 18-912, AMP 20-1014), the Västerbotten regional county, and from Karin and Krister Olsson. The computations were enabled by resources provided by the Swedish National Infrastructure for Computing (SNIC) at the High Performance Computing Center North (HPC2N) in Umeå, Sweden, partially funded by the Swedish Research Council through grant agreement no. 2018-05973.

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

## Appendix A. Related Works

The are many causes of the bias field. Most of it can be attributed to local flip angle variations caused by a non-uniformity in the $B_0$ static field and the transmitted $B_1$ field (McRobbie et al., 2006), to tissue-specific radio frequency penetration (Belaroussi et al., 2006) and to heterogeneous receive $B_1$ fields. Vendors frequently have built-in solutions that correct for the non-uniform receiver-coils—*e.g.* PURE for GE Healthcare (Chicago, Illinois, United States) and Prescan Normalization for Siemens (Erlangen, Germany).

For retrospective bias field correction, one of the most popular methods is an iterative, high-frequency content maximization method proposed (Sled et al., 1998), called N3. Their extensive research on the characteristics of the bias fields provides a strong theoretical foundation for the correction. The method is subject to constant renewals and modifications (Lin et al., 2011; Larsen et al., 2014), of which one of the most popular is N4ITK (Tustison et al., 2010). The N4ITK is often used in clinical practice due to its high accuracy. This performance comes at the cost of execution time, and a list of parameters to be tuned.

Despite its complex combination of origins, a bias field can be described as a low-frequency multiplicative imaging artefact causing a smooth intensity variation spatially across the image. This is an important characteristic, as correctors might approximate the field using a Gaussian (Wells et al., 1994) or a quadratic estimation (Dawant et al., 1993), or B-splines for the case of N4ITK, smoothness is nonetheless often assumed and exploited in different methods (Sled et al., 1998).

Deep learning has been applied to bias field correction before, most often on a particular anatomy or contrast (Sridhara et al., 2021; Goldfryd et al., 2021) without the main aim of producing a generalized model. They are also often combined with another medical imaging task, such as tissue segmentation (Fengkai et al., 2019; Zhang and Song, 2019) or signal reconstruction (Gaillochet et al., 2020), which keeps the method limited to that particular context. The authors are not aware of any other work that aims to achieve such high generalization for the bias field correction of MRI data using deep learning.

## Appendix B. Example from the phantom dataset

| Input Image | Ground Truth | N4ITK | Explicit | Implicit |
|---|---|---|---|---|

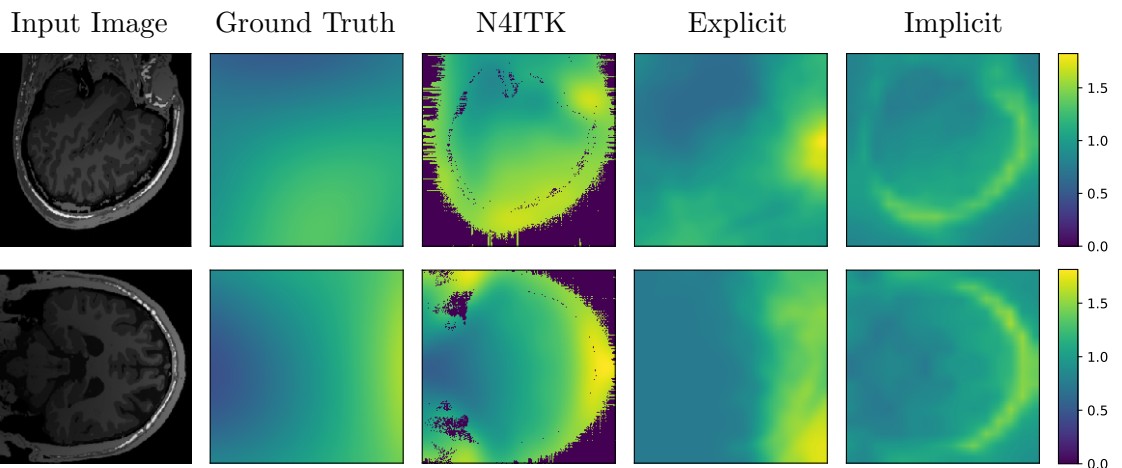

Figure 3: Two examples from the testing BrainWeb dataset for the recovered bias fields from the three evaluated methods. All bias fields are on the same scale.

## Appendix C. Examples from the real dataset

| Original | N4ITK | Explicit | Implicit |

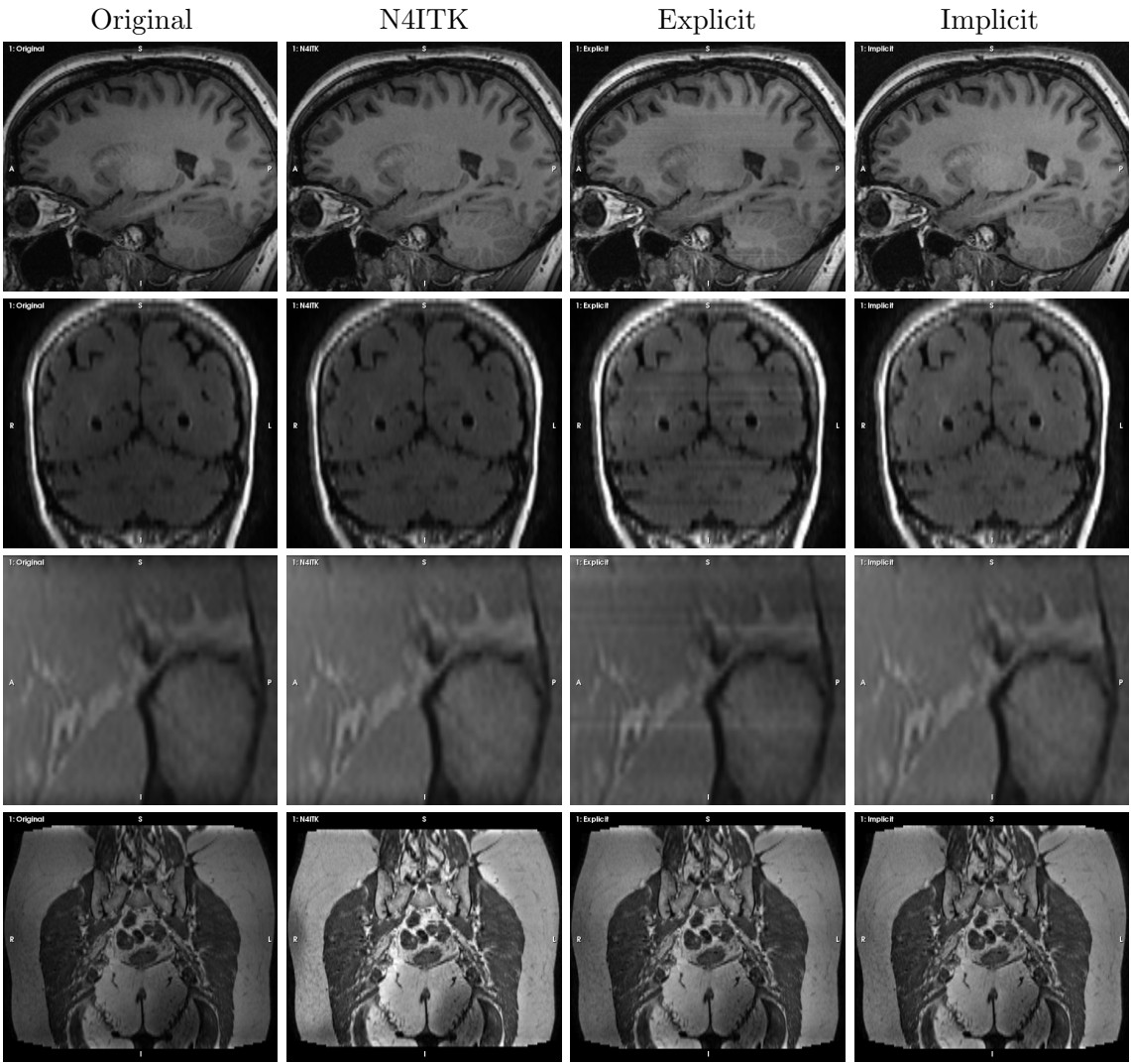

Figure 4: Four examples from the real datasets, and their corrections, all using the same scaling. The presented slices are orthogonal to the correction axis of the trained models. The images were corrected volume-wise for N4ITK. The presented datasets from top to bottom are: $T_1$-weighted and $T_2$-weighted brain scans, LAVA-FLEX and $T_1$-weighted pelvic scans.

