# OpenReview forum: "MRI bias field correction with an implicitly trained CNN."
_MIDL.io/2022/Conference — MIDL 2022_

### Official Review · Reviewer_3f6v · 2022-01-18

**Confidence:** 4
**Preliminary Rating:** 3
**Recommendation:** Poster

**Summary:**

The authors proposed an implicit (i.e., self-supervised) method for training deep learning models for MRI bias field correction. Their proposal allows training the bias field correction model using non-MRI images.

The authors compare their proposal against supervised training and the commonly used N4ITK techniques. Their results seem to be on par with N4ITK results with the advantage of it being considerably faster.



**Strengths:**

The proposed method for implicitly training an MRI bias field correction network is simple, mathematically sound, and it allows the usage of natural images (i.e., not MRI) during the training stage. The authors did not do any neural architecture search but using a simple U-net architecture, they obtained results comparable to N4ITK with the advantage of being nearly 100 times faster (running times were compared using CPU).

**Weaknesses:**

In my opinion, there were a few factors that reduced the quality of the paper. I list them below.

1. The literature review seems quite limited. The most recent reference is from 2016 and is not related to bias field correction. The papers related to bias field correction seem to be > 10 years old. A quick search on Google Scholar indicated more recent works in the field that leverage deep learning. A better literature review would help to better put the authors' work in context.

2. Some experimental design choices seemed a bit arbitrary and could potentially have been discussed in the paper. For example, the implicit model was trained using ImageNet data, but for validation (i.e., model selection) the authors used simulated data from BrainWeb. Why BrainWeb and not ImageNet data was used for the implicit model selection? Does this affect the results in any way?

**Deanonymize Review:**

no

**Detailed Comments:**

I list a few minor comments in this section.

1. The references formatting should probably be standardized. Some papers are written in lowercase and others in uppercase. The year of reference number 6 is missing.

2. The authors refer to figure 2 before referring to figure 1.

3. Why do the authors show the corrections orthogonal to the correction axis? (is it a correction axis or correction plane?) I imagine corrections are done slice-by-slice)

4. I imagine the authors are reporting average results for CV and SSIM. I believe the standard deviation should also be reported.

5. Many of the differences in the results show up only in the third decimal house of the metrics. Did the authors test for statistical significance?

**Final Rating After The Rebuttal:**

3: Borderline

**Justification Of The Final Rating:**

The authors responded to some of my comments. My biggest concern with the paper is still the fact that the main comparison is against something that was proposed 12 years ago, which is still largely used but may not represent the current state-of-the-art.

**Paper Type:**

methodological development

**Questions To Address In The Rebuttal:**

1. Please explain why the implicit method uses the BrainWeb data as the validation set and not the ImageNet data.

2. Why the authors didn't include more recent works in the literature review?

3. Why do the authors show the corrections orthogonal to the correction axis? (is it a correction axis or correction plane? I imagine corrections are done slice-by-slice)

4. Why the standard deviation of the results was not reported?

5. Why the authors did not test the results for statistical significance?

**Special Issue:**

no

---

### Official Review · Reviewer_fDkV · 2022-01-19

**Confidence:** 3
**Preliminary Rating:** 5
**Recommendation:** Oral

**Summary:**

The authors propose a method for training a network to perform bias-field correction. The method does not require ground-truth bias field maps for training, but rather uses a corrupted image and the corrupted image with a further simulated bias field, and constrains the ratio of these images with bias-field correction applied to be equal to the simulated biased field. The authors validate on simulated, phantom, and real data, comparing to the traditional correction tool N4ITK. The author's model achieves similar results, and faster runtimes.

**Strengths:**

- The idea is, to my knowledge, novel and explained clearly
- The authors show their method can be trained on natural images, e.g. ImageNet, and works on medical images, which is interesting
- The paper is well validated on simulated, phantom and real datasets
- The trained model is made available

**Weaknesses:**

- All the work is carried out in 2D. Realistically a bias-field correction method would need to be 3D to avoid slice-wise artefacts and it would have been nice to see some 3D results.

- Much of the work relies on images simulated using brainweb. This is a very simple simulation model, which solves the Bloch equations once for each tissue type and sums the results according to a voxel map. I think the evaluation would be improved by making use of a more realistic simulator; but having said that the inclusion of results on phantom and real data makes this weakness less important.

**Deanonymize Review:**

yes

**Detailed Comments:**

- One of the big advantages of simulated data is that a ground truth, in this case the true bias field, is readily available. Why did you  not report the difference between the real and estimated bias field for simulated data? Instead you relied on surrogate metrics such as SSIM.

- I thought it interesting that you were able to train youe model on imagenet and show it works on medical image data, but was left wondering if the method would perform better if trained on medical imaging data. Perhaps a more extended version of this work such as a journal paper could include these results?

- I would be interested to know how sensitive the performance of your model is to the parameters chosen for the generation of the simulated bias fields. This has implications for the use of your method to correct other artefacts that might be harder to simulate faithfully.

**Final Rating After The Rebuttal:**

5: Strong Accept

**Justification Of The Final Rating:**

I'm satisfied with the author's reponse. In particular, despite the method being 2D it does seem from the results and Figure 4 that the method does not cause slicewise correction artefacts when viewed in an orthogonal plane. I am leaving my recommendation unchanged.

**Paper Type:**

methodological development

**Questions To Address In The Rebuttal:**

- Why did you not implement and test your method in 3D?
- Could you do better by training the implicit model on medical data?
- How sensitive is your method to the quality of the simulated bias fields?

**Special Issue:**

yes

---

### Meta-Review · Area_Chair_ANCt · 2022-02-14

**Recommendation:** Accept (Poster)
**Confidence:** 4

**Metareview:**

The idea presenting in this paper is interesting and simple, and provides promising preliminary results. The experiment framework is well-designed and the novelty is good enough. Moreover, the trained model is made available. Consequently, I would recommend a acceptation with poster presentation.

---

### Decision · Program_Chairs · 2022-02-28

Accept